# Endolichenic Fungi: A Promising Medicinal Microbial Resource to Discover Bioactive Natural Molecules—An Update

**DOI:** 10.3390/jof10020099

**Published:** 2024-01-25

**Authors:** Wenge Zhang, Qian Ran, Hehe Li, Hongxiang Lou

**Affiliations:** Department of Natural Products Chemistry, Key Lab of Chemical Biology (MOE), School of Pharmaceutical Sciences, Shandong University, No. 44 West Wenhua Road, Jinan 250012, China; 202221066@mail.sdu.edu.cn (W.Z.); 202236871@mail.sdu.edu.cn (Q.R.); 202336801@mail.sdu.edu.cn (H.L.)

**Keywords:** lichens, endolichenic fungi, secondary metabolites, biological activity

## Abstract

Lichens are some of the most unique fungi and are naturally encountered as symbiotic biological organisms that usually consist of fungal partners (mycobionts) and photosynthetic organisms (green algae and cyanobacteria). Due to their distinctive growth environments, including hot deserts, rocky coasts, Arctic tundra, toxic slag piles, etc., they produce a variety of biologically meaningful and structurally novel secondary metabolites to resist external environmental stresses. The endofungi that live in and coevolve with lichens can also generate abundant secondary metabolites with novel structures, diverse skeletons, and intriguing bioactivities due to their mutualistic symbiosis with hosts, and they have been considered as strategically significant medicinal microresources for the discovery of pharmaceutical lead compounds in the medicinal industry. They are also of great importance in the fundamental research field of natural product chemistry. In this work, we conducted a comprehensive review and systematic evaluation of the secondary metabolites of endolichenic fungi regarding their origin, distribution, structural characteristics, and biological activity, as well as recent advances in their medicinal applications, by summarizing research achievements since 2015. Moreover, the current research status and future research trends regarding their chemical components are discussed and predicted. A systematic review covering the fundamental chemical research advances and pharmaceutical potential of the secondary metabolites from endolichenic fungi is urgently required to facilitate our better understanding, and this review could also serve as a critical reference to provide valuable insights for the future research and promotion of natural products from endolichenic fungi.

## 1. Introduction

As one of the most critically important groups of unique fungi, lichens are broadly acknowledged for their unique biological characteristics, lacking any obvious differentiation of their roots, stems, and leaves. Lichens are naturally occurring, stable microorganisms, and they are mutually beneficial organic complexes with a complex composition of fungal and chlorophyll organisms [1,2]. Mutualistic symbiosis is the biologically distinctive characteristic that allows lichens to be differentiated from other common plants [3]. Among the symbiotic microorganisms existing in lichens, ascomycetes are the pivotal symbiotic fungi that account for their fungi diversity. Basidiomycetes are also important members of endolichenic microorganisms, and symbiotic partners are usually present as cyanobacteria or green algae [4].

Lichens can survive in a variety of extreme living environments, such as during nutrient shortages, under wet, cold, and dark conditions, in high temperatures. Notably, they often grow on the surfaces of rocks or trees, without any typical nutrient requirements, and they are widely distributed across almost all land surfaces on Earth, even when there are no suitable loamy soils for the survival of other plants or animals. It is well known that lichens can be readily encountered in nature, ranging from the North and South Poles to the equator, from mountains to plains, forests to deserts, and land to sea [5,6,7,8,9]. Due to their complex environmental living conditions, they display an outstanding capacity to produce numerous intriguing bioactive secondary metabolites to resist external habitat stresses [3].

Throughout the centuries-long use of traditional medicine to treat human health conditions, lichens have been widely used in the pharmacopeias of different countries as effective remedies for various diseases; thus, they have attracted attention from both pharmaceutical and natural product chemists, who have investigated their bioactive chemical compositions in recent decades [10,11]. The critical chemical constituents of lichens have been consequently demonstrated, and they are usually exemplified by depsides and their esters, depsidones, polysubstituted benzenes, anthraquinones, dibenzofurans, terpenoids, and steroids, together with lichen polysaccharides [12]. These structurally diverse natural products from lichens have been illustrated to show potent enzyme-inhibitory, antibacterial, anti-fungal, anti-viral, anti-tumor, insecticidal, antioxidant, and other biological properties, which closely correspond to their traditional medicinal applications and therapeutic efficacy [4,13]. Notably, numerous excellent scientific efforts to elucidate the bioactive chemical composition of lichens have disclosed that different species of lichens can produce various types of bioactive natural compounds with different structural characteristics [14].

Fungi are well known as a promising strategic pharmaceutical bioresource for the discovery of structurally intriguing and biologically significant medicinal lead compounds, and they show great advantages with the pivotal characteristics of a short growth cycle, easy fermentation, and readily operational cultivation. The number of natural products derived from fungi and the proportion of active substances in all natural sources are sharply increasing year by year, accompanied by technological innovations of natural product chemistry towards compound separation and structure identification in recent decades. Moreover, fungi play important roles in plants, terrestrial ecosystems, and other related ecosystems. Fungi found in unique habitats have shown great potential in pharmaceutical research and innovative drug development due to their unique metabolic and defense systems under the influence of complex environmental factors, and the biologically meaningful natural products derived from fungi material serve as an important source of drug lead compounds in both the medicinal and medical industries.

As a plant group closely related to fungi, lichens are well known to be composed of a variety of fungi, such as surface symbiosis fungi and endophytic fungi [15,16]. Endolichenic fungi (ELF) are usually recognized as the fungi living inside healthy lichen tissues, and they do not cause any directly or indirectly noticeable negative effects on the lichen thallus [17]. Conventionally, lichens supply stable living environments and suitable nutrients to meet the survival requirements of endolichenic fungi for normal growth. In turn, ELF will produce abundant secondary metabolites for lichens to resist biotic stresses due to extreme environments and accelerate their growth. ELF are similar to plant endophytic fungi; they also show a potent ability to produce pharmacologically meaningful secondary metabolites with novel structures and extensive biofunctions due to their unique living environments. They are thus becoming a new research hotspot for the discovery of new active compounds [17].

The first chemical research study on the secondary metabolite system of ELF can be traced back to 2007 [18]. As a result, Paranagama successfully reported the isolation and purification of two new heptaketides from ELF, and their cytotoxicity was also evaluated. This excellent research effort was the first scientific report on the endophytic fungal metabolites of lichens, and it marked the beginning of extensive chemical research in the field of natural product chemistry using ELF. Since then, a growing number of pharmaceutical and natural product chemists have carried out fundamental studies and innovative drug developments on the metabolites of lichens and endophytic fungi. To date, thousands of secondary metabolites with different structural types, including alkaloids, steroids, xanthones, benzopyranoids, peptides, and allycylic compounds, have been discovered from ELF [19]. Moreover, a huge number of these natural compounds from ELF have exhibited significant biological activity, such as antibacterial, antioxidant, cytotoxic, reversal resistance, and enzyme inhibitory activities [20].

The chemical diversity and biological activity of the secondary metabolites of ELF were comprehensively reviewed by Gao et al., covering the period up to 2015 [19]. Since then, efforts towards the discovery and pharmacological investigation of the secondary metabolites of ELF have rapidly increased. According to the literature statistics from the database of Web of Science, there have been more than 90 articles on the chemistry of the secondary metabolites of endolichenic fungi, and up to 34 species of endophytic fungi in lichens have been studied since the first work on the secondary metabolites of endolichenic fungi by Paranagama et al. In the previous reports on ELF, a total of 583 natural products have been reported from endolichenic fungi. Among them, 370 compounds are novel ones, including polyketides, polyphenyls, terpenoids, steroids, alkaloids, and others. Notably, many more related studies have described new discoveries regarding the secondary metabolites of ELF in other journals on natural product chemistry, thus highlighting the relevance of ELF research.

The growing number of outstanding research works and rapid developments in the field of bioactive natural products from ELF make them increasingly important in the modern medicinal industry. Therefore, our present overview intends to provide relevant scientific information covering the recent progress in the study of bioactive natural products from ELF for readers from both the fundamental medicinal research and related industries. Previous reviews of bioactive natural products from ELF have focused on their chemical compositions and pharmacological effects, but they fail to provide an extensive discussion and analysis of endolichenic fungi metabolites in terms of their research status and future trends. In this review, the structural characteristics and composition types, biological activities, potential applications, strain distributions, and origins, as well as the research status and future trends of the bioactive secondary metabolites from ELF are comprehensively summarized, discussed, and analyzed, covering the period since 2015, in order to provide a prospective view to address the increasing demand for ELF-related fundamental research and drug innovation.

## 2. Different Types of Natural Products from Endolichenic Fungi

In recent years, chemical efforts devoted to the discovery of pharmacologically meaningful secondary metabolites of ELF have emerged and seen massive growth. Since 2015, a total of 519 natural products, excluding the repeatedly isolated ones, have been reported from ELF. Among them, 266 are novel compounds, including polyketides, polyphenyls, terpenoids, steroids, alkaloids, and others. According to a statistical analysis (Figure 1) of the structural types of natural products and their numbers from ELF, it is found that polyketides account for the largest proportion of structural types discovered from ELF each year. Overall, polyketides are the most abundant members among the secondary metabolites of ELF, followed by alkaloids, terpenes, steroids, and other types of natural products. In addition, the total number of compounds discovered each year remained stable, ranging from 40 to 80, with the exception of 2022.

### 2.1. Polyketides

Polyketide compounds account for the most critical family of secondary metabolites of ELF. This intriguing result might be due to their specific biosynthetic gene clusters. Polyketides are one of the most important groups of natural products, and they exhibit a variety of biologically meaningful properties. There are a variety of natural products belonging to polyketides, such as quinones, anthrones, chromogenones, isocoumarins, and other small molecules. Since the first anthraquinones from endolichenic fungi were reported by Paranagama et al. in 2007 [18], 419 members of the polyketide family have been consequently reported so far, of which 258 natural polyketides have been reported since 2015. In this review, the polyketides are further subdivided into simple aromatic polyketides, complex aromatic polyketides, and non-aromatic polyketides. Their chemical structures and biological activities are also extensively summarized and discussed as follows.

#### 2.1.1. Simple Aromatic Polyketides 

Quinones are organic compounds characterized by a typical six-membered cyclic diketone structure with two double bonds, which are formulated as a highly conjugated scaffold. Among these quinone compounds, their chemical structure types can be divided into benzoquinones, anthraquinones, perylene quinones, and quinone derivatives. Notably, benzoquinones are the most abundant family, and they show fascinating structural diversity. Quinones frequently occur in ELF and usually show potent biological activity. To date, 29 quinones (Figure 2) have been successfully discovered from ELF since 2015. In this part, the structural characteristics and biological activities of the quinones reported since 2015 are summarized and discussed.

In 2016, endocrocin (**1**) was isolated from an EtOAc extract of the endolichenic fungal strain *Sporormiella irregularis* 71-11-4-1 by Yang et al. [21]. In the same year, Zhou et al. isolated 13 anthraquinone compounds from the endolichenic fungus *Biatriospora* sp. 8331C [22], and the new compounds biatriosporins G-L (**2**–**7**) were reported for the first time in nature. The other anthraquinones were identified as 6-O-demethyl-5-deoxyfusarubin (**8**), pyranonaphthoquinone (**9**), 6-deoxy-7-O-demethyl-3,4-anhydrofusarubin (**10**), 6-deoxy-3,4-anhydrofusarubin (**11**), ascomycone A (**12**), ZSU-H85 A (**13**), and 3-acetyl-2,8-dihydroxy-6-methoxy anthraquinone (**14**). In 2017, Wang et al. reported two other anthraquinones, which were 6-O-demethylbostrycin (**15**) and bostrycin (**16**) [23]. Notably, **16** exhibited significant cytotoxicity against the L5178 murine lymphoma cell line with an IC_50_ value of 1.7 μM.

Moreover, thirteen benzoquinone derivatives (**17**–**28**) were isolated from the lichen endophytic fungus *Ulospora bilgramii* 8344B in 2020 [24]. Among them, ulosporins A-G (**17**–**23b**) were reported as new compounds for the first time. The other six benzoquinone compounds, 2-acetonyl-3-methyl-5-hydroxy-7-methoxynaphthazarin (**24**), 6-deoxy-3,4-anhydrofusarubin (**25**), ascomycone A (**26**), ascomycone B (**27**), and 6-deoxybostrycoidin (**28**) were revealed as known compounds. A bioactivity assay test showed that ulosporin G (**23**) significantly inhibited the growth of the human cancer cell lines A549, MCF-7, and KB tumor cells, with IC_50_ values of 1.3, 1.3, and 3.0 μM, respectively. Furthermore, compound **23** caused apoptosis in A549 cells by arresting the G0/G1 cell cycle and inducing DNA damage. In 2023, Varlı et al. isolated an anthraquinone, 1′-O-methyl-averantin (**29**), from the endolichenic fungus *Jackrogersella* sp. EL001672 via the bioactivity-guided fractionation method [25]. Similarly, compound **29** displayed weak antioxidant activity but significant cytotoxicity against the CSC221, CaCo2, DLD1, and HCT116 cancer cell lines, with IC_50_ values ranging from 18.35 to 22.78 µg/mL. An anticancer mechanism study demonstrated that compound **29** suppressed cancer stemness via Sonic hedgehog and Notch signaling.

Xanthones are also a very important type of natural product of ELF. A total of 54 xanthones have been reported from ELF, and 14 of them (Figure 3) were discovered after 2015. In the molecular structure of xanthones, a ketone group is located at the edge of an anthracene ring. This unique structure gives xanthone unique chemical and physical properties, as well as various types of biological activities, including antihypertensive, anticonvulsive, antithrombotic, and antitumor activities. In the endolichenic fungal strain *Sporormiella irregularis* No. 71-11-4-1, Yang et al. also obtained a new xanthone glycoside, sporormielloside (**30**), and a known xanthone (**31**) in 2016 [21]. In 2017, six xanthone derivatives, including the new 8-hydroxy-3-hydroxymethyl-9-oxo-9H-xanthene-1-carboxylic acid methyl ethe (**32**) and the known xanthones 8-hydroxy-3-methyl-9-oxo-9H-xan-thene-1-carboxylic acid methyl ether (**33**), norlichexanthone (**34**), anomalin A (**35**), anomalin B (**36**), and sydowinin B (**37**) were reported by Zhou et al. [22]. 

Padhi et al. reported the isolation and identification of three xanthones, funiculosone (**38**), mangrovamide J (**39**), and ravenilin (**40**), from a crude extract of the endolichenic fungus *Talaromyces funiculosus* in 2019 [26]. Among them, compound **38** was reported as an undescribed substituted dihydroxanthene-1,9-dione that showed moderate antibacterial activity, with MIC values ranging from 23 to 58 μg/mL towards *E. coli* and *Staphylococcus aureus*. Moreover, **38** displayed significant inhibitory activity against *Candida albicans*, with an MIC value of 35 μg/mL. The daldipyrones A–C (**41**–**43**) were identified from an endolicanic fungus, *Daldinia pyrenaica* 047188, by Lee et al. in 2023 [27]. Notably, the structures of daldipyrones A–C (**41**–**43**) were determined to share an unprecedented caged xanthone [6,6,6,6,6] polyketide future with a spiro-azaphilone unit using a spectroscopic analysis and chemical derivatization. Interestingly, the biological evaluation results illustrated that daldipyrenone A (**41**) displayed conspicuous antimelanogenic activity with an EC_50_ value lower than that of the positive control as well as moderate adiponectin-secretion-promoting activity.

According to the structural characteristics of these simple aromatic polyketide compounds, their possible biosynthetic pathways were proposed, as shown in Figure 1 [21]. The simple aromatic polyketides from ELF were considered to be genenrated from a commutual precusor, C_16_-octaketide. Firstly, C_16_-octaketide was catalyzed by non-reducing polyketide synthase to undergo aldol condensation and cyclization to yield the intermediate atrochryone carboxylic acid, which can be further oxidized and dehydrated to generate endocrocin (**1**) or subjected to a series of chemical transformations involving dehydration/decarboxylation/spontaneous oxidation to offer the critical compound emodin. Emodin can be transformed into intermediate **a** through the enzymatically catalyzed oxidative ring-opening reaction with a carbon reduction between C-4 and C-5. Moreover, emodin can also be subsequently dehydrated to generate the intermediate **b**. Compound **31** is derived from intermediate **b** via methylation. Intermediate **b** can transform into intermediate **c** though decarboxylation and oxidation reactions, whereas methylation and glycosylation can attain compound **30** from intermediate **c**. 

In addition, chromones and isocoumarins are also widely observed and recognized as very common secondary metabolites (Figure 4) in endolichenic fungi. Since 2015, 25 chromogenic ketones together with 22 isocoumarins have been reported. For example, Kim et al. isolated two new chromones, phomalichenones C-D (**44**–**45**), and three known chromone derivatives (**46**–**48**) from the endolichenic fungus *Phoma* sp. in 2018 [28]. In 2019, a phomalone derivative, (7-hydroxy-8-[2-hydroxyethyl]-5-methoxy-2-methylchro-man-4-one (**58**), was isolated from the endolichenic fungus *Cochliobolus kusanoi* [29]. In the same year, two other new chromone derivatives, ophiochromanone (**49**) and ophiolactone (**50**), were isolated and identified from the EtOAc extract of *Ophiosphaerella korrae* by Li et al. [30]. Furthermore, the chromones spororrminone A (**51**) and 2-epi-spororrminone A (**52**) were also isolated as two new compounds from the crude extract of the endolichenic fungus *Sporormiella irregularis*; they represent the first examples of 2-(5-oxotetrahydrofuran-2-yl)chromone with a 7-carboxylic functional group [31]. However, neither of these two new compounds showed obvious antimicrobial activity or cytotoxicity.

In 2020, an investigation of the endolichenic fungus *Daldinia eschscholzii* successfully led to the discovery of one known polyketide, 5-hydroxy-2-methylchroman-4-one(**53**) [32]. In 2021, Zhang et al. isolated two new chromone derivatives, (5*R*,7*R*)-5,7-dihydroxy-2-methyl-5,6,7,8-tetrahydro-4H-chromen-4-one (**54**) and (5*R*,7*R*)-5,7-dihydroxy-2-propyl-5,6,7,8-tetrahydro-4H-chromen-4-one (**55**), together with two known chromone compounds, (5*R*,7*S*)-5,7-dihydroxy-2-methyl-5,6,7,8-tetrahydro-4H-chromen-4-one (**56**) and (5*R*,7*S*)-5,7-dihydroxy-2-propyl-5,6,7,8-tetrahydro-4H-chromen-4-one (**57**), from the endolichenic fungus *Daldinia* sp. CPCC 400770 [33]. Their bioactivity tests indicated that compounds **54** and **56** displayed obvious anti-influenza A virus (IAV) activity, with IC_50_ values of 16.1 and 9.0 mM, respectively.

Zhou et al. obtained a series of ramulosin derivatives (Figure 5), including biatriosporin M (**59**), 5-hydroxymellein (**60**), 4-hydroxymellein (**61**), 3,4-dihydro-4,5,8-trihydroxy-3-methylisocoumarin (**62**), and 5-hydroxyramulosin (**63**), from the endolichenic fungus *Biatriospora* sp. 8331C in 2016 [22]. In 2017, six isocoumarins, decarboxycitrinone (**64**), 6,8-dihydroxy-4-hydroxymethyl-3,5-dimethyl-isochro-men-1-one (**65**), decarboxyhydroxycitrinone (**66**), acremonone G (**67**), O-methylmellein (**68**), and *trans*-4-hydroxymellein (**69**), were isolated by Wang et al. from the endolichenic fungus *Apiospora montagnei* [23]. Among them, compound **67** exhibited significant cytotoxicity against the L5178 murine lymphoma cell line, with an IC_50_ value of 2.7 μM. The isocoumarins (3*R*,8*S*)-dihydroxy-3-hydroxymethyl-6-methoxy-4,5-dimethylisochroman-1-one (**70**), ophioisocoumarin (**71**), and (*R*)-3,4-dihydro-4,8-dihydroxy-6-methoxy-4,5-dimethyl-3-methyleneisochromen-1-one (**72**) were also isolated from the endolichenic fungus *O. korrae* [30]. 

Shevkar et al. obtained a new compound, peniazaphilin B (**73**), from the endolichenic fungus *Talaromyces pinophilu* in 2022 [34]. In the same year, Yuan et al. isolated two new 3,4-dihydroisocoumarin derivatives with a novel dihydrothiophene skeleton from *Talaromyces* sp. and named them as talarolactone A (**74**) and talarolactone A (**74a**) [35]. Moreover, a plausible biosynthetic pathway for **74** has been proposed, as shown in Figure 2. The isocoumarin skeleton is speculated to be formed through the polyketide synthetase (PKS) pathway, whereas the dihydrothiophene scaffold might be biosynthetically constructed by a cascade reaction of sulfhydrylation and cyclodehydration. In 2023, three new isocoumarin analogues, aspermarolides A–C (**75**–**77**), coupled with two known related analogues, 8-methoxyldiaporthin (**78**) and diaporthin (**79**), were isolated from a culture extract of *Aspergillus flavus* CPCC 400810 [36]. However, none of these showed notable cytotoxic activity against the HepG2 and Hela cell lines.

In addition to the aforementioned classical aromatic polyketide compounds, there are many other related aromatic polyketides that have been revealed to exist in ELF (Figure 6). For example, Samanthi et al. reported a new macrocyclic ketone, 5-methoxy-4,8,15-trimethyl-3,7-dioxo-1,3,7,8,9,10,11,12,13,14,15,15*α*-dodecahydrocyclododeca[de]isochromene-15-carboxylic acid (**80**), in an endolichenic fungus, *Curvularia trifolii*, in 2015 [37]. The bioactivity screening of **80** was performed using a DPPH antioxidant assay, which thus showed that **80** exhibited radical scavenging activity with an IC_50_ value of 1.3 ± 0.2 mg/mL. Moreover, **80** showed significant anti-inflammatory activity comparable to that of the standard anti-inflammatory drug aspirin. In 2016, Yuan et al. obtained two new polyketides, myxotritones B–C (**81**–**82**), from the endolichenic fungus *Myxotrichum* sp., using the OMSAC (one strain, many compounds) method [38]. Moreover, the glycine *N*-(2,3-dihydroxybenzoyl)-methyl (**83**) was disclosed from an endolichenic fungus, *Tolypocladium* sp. 4259a, in 2017 by Hu et al. [39].

In 2018, Kim et al. isolated four new phomalone derivatives, phomalichenones A–B (**84**–**85**), and four known compounds, (2,4-dihydroxy-3-(2-hydroxyethyl)-6-methoxyphenyl)-3-hydro-xybutan-1-one (**86**), (*E*)-1-(2,4-dihydroxy-3-(2-hydroxyethyl)-6-methoxyphenyl)but-2-en-1-one (**87**), phomalone (**88**), and deoxyphomalone (**89**), from the endolichenic fungus *Phoma* sp. [28] The cytotoxic and anti-inflammatory activities of all the compounds were determined, and it was found that compounds **84** and **87** showed significant anti-inflammatory activity, with IC_50_ values of 9.4 ± 0.5 and 7.4 ± 2.8 μM, respectively. Moreover, the authors speculated that the presence of a double bond on the side chain in these phomalone derivatives may be essential for their inhibitory effect against NO production.

In 2019, two known compounds, (*R*)-7-hydroxy-3-((*S*)-1-hydroxyethyl)-5-methoxy-3,4-dime-thylisobenzofuran-1(3H)-one (**90**) and clearanol E (**91**), were isolated from the EtOAc extract of *O. korrae* by Li et al. [30]. In the same year, Yang et al. reported two phomalone derivatives, 1-(2,4-dihydroxy-3-[2-hydroxyethyl]-6-methoxyphenyl)butan-1-one (**92**) and 1-(2,4-dihydroxy-3-[2-hydroxyethyl]-6-methoxyphenyl)-3-hydroxybutan-1-one (**93**), which were isolated from the endolichenic fungus *Cochliobolus kusanoi* [29]. Compound **92** was a new compound, but it was easily shown that compounds **92**–**93** and **58** can be converted into one another under suitable reaction conditions. For example, **92** is hydroxylated at the C-3 position through a Michael addition reaction with a molecule of water to contribute to compound **93**. Similarly, compound **58** might also be formed by the dehydration of the hydroxyl groups of **93** between the C-3 and C-2′ enablers.

In 2020, Padhi et al. isolated a new 6-benzyl-c-pyrone, aspergyllone (**94**), and a known 6-benzyl-c-pyrone, carbonarone A (**95**), from the secondary metabolites of the endolichenic fungus *Aspergillus niger* [40]. The bioactive tests suggested that **94** displayed strong antimicrobial activity, with IC_50_ values ranging from 35 to 97 µg/mL. Interestingly, **94** showed strong selective antifungal activity against *Candida parapsilosis* (Ashford), with an IC_50_ value of 52 µg/mL. In the same year, a chemical investigation of the endolichenic fungus *D. eschscholzii* led to the discovery of one known polyketide, 8-methoxynaphthalen-1-ol (**96**) [32]. Compound **96** showed strong radical scavenging ability in a DPPH assay, with an IC_50_ value of 10.2 ± 5.8 µg/mL, which was much higher than that of the standard drug butylated hydroxy toluene (BHT). Another excellent investigation of the T-DNA insertion transformant (strain TR-74) of *X. grammica* KCTC 13121BP led to the discovery of a novel benzoquinone, 2-(hydroxymethyl)cyclohexa-2,5-diene-1,4-dione (**112**), and its derivative, 2,5-dihydroxybenzaldehyde (**113**) [41].

In 2021, Zhang et al. isolated three new phenolic compounds, daldispols A-C (**97**–**99**), and five known phenolic compounds, 2-phenylethyl-*β*-D-glucopyranoside (**100**), stachyline C (**101**), 3-methoxy-4-hydroxy-phenylethanol (**102**), 3-hydroxy-4-methoxy-phenylethanol (**103**), and *p*-hydroxyphenethyl alcohol (**104**), from the endolichenic fungus *Daldinia* sp. CPCC 400770 [33]. Their bioactivity experiments indicated that compounds **97**, **98**, and **100** displayed obvious anti-influenza A virus (IAV) activity, with IC_50_ values of 12.7, 6.4, and 12.5 μM, respectively. Moreover, compound **104** exhibited anti-ZIKV activity, with an inhibitory ratio of 42.7% at 10 μM. In the same year, the chemical investigation of an endolichenic *Aspergillus chevalieri* led to the discovery of seven C7-alkylated salicylaldehyde derivatives, asperglaucins A-B (**105**–**106**), tetrahydroauroglaucin (**107**), flavoglaucin (**108**), 2-(1′,5′-heptadienyl)-3,6-dihydroxy-5-(3″-methyl-2″-butenyl)benzaldehyde (**109**), isodihydroauroglaucin (**110**), and 2-(E-3-heptenyl)-3,6-dihydroxy-5-(3-methyl-2-butenyl)-benzalde-hyde (**111**) [42]. It should be noted that asperglaucins A–B (**105**–**106**) were new compounds reported in nature for the first time, and they displayed significant antibacterial activity against *Pseudomonas syringae pv actinidae* (*Psa*) and *Bacillus cereus*, with an MIC value of 6.25 μM. The antibacterial mechanism study revealed that **105**–**106** played an antibacterial role by changing the external structures of *B. cereus* and *Psa*, which thus caused the rupture or deformation of the cell membrane to kill bacterial cells. Furthermore, phenolics **107**–**109** were shown to display obvious antioxidative effects. 

In 2023, Gamage et al. found that the strain *Arthrinium* sp. EL000127 could produce phthalide derivatives, such as the known compound 3-O-methylcyclopolic acid (**114**) and two new analogues, 3-O-phenylethylcyclopolic acid (**115**) and 3-O-*p*-hydroxyphenylethylcy-clopolic acid (**116**) [43]. The compounds **114**–**116** exhibited very weak cytotoxicity against HUVECs, with IC_50_ values of 215.6, 43.8, and 1.8 mM, respectively. Moreover, compound **116** exhibited antiangiogenic activity by inhibiting the mRNA expression of genes to regulate epithelial cell survival and motility, suggesting that compound **116** is a potent antiangiogenic agent with promising potential as a lead compound for the development of novel cancer therapeutic agents. 

In the same year, a novel compound, neurosporalol L (**117**), was obtained from the endolichenic fungus *Neurospora ugadawe*, which was isolated from the lichen host *Graphis tsunodae* Zahlbr [44]. The active results showed that **117** displayed obvious antioxidant activity. Notably, the antioxidant activity of **117** was stronger than that of the positive control BHT, and its IC_50_ value was low: 3.48 ± 0.33 µg/mL. In 2022, the compound 4,6-dihydroxy-5-methylphthalide (**118**) was isolated from the endolichenic fungus *Talaromyces* sp., associated with *Xanthoparmelia angustiphylla*, by Yuan et al. [35]. Kim et al. developed an efficient methodology using a feature-based MS/MS molecular networking analysis to discover the novel secondary metabolites from the reported endolichenic fungus, and this successfully resulted in the isolation of four compounds, 1,8-dimethoxy naphthalene (**119**), 8-methoxy-1-naphthol (**120**), 1-(2,6-dihydroxyphenyl)butan-1-one (**121**), and 2,6-dihydroxya-cetophenone (**122**), from *Daldinia childae* in 2023 [45].

#### 2.1.2. Simple Nonaromatic Polyketides

Generally, nonaromatic polyketides are recognized as polyketides without unambiguous aromatic ring systems or aromatic substituents. As summarized in the review, the nonaromatic polyketides isolated from endolichenic fungi since 2015 include furanones, cyclopentanones (Figure 7), pyranones (Figure 8), cyclohexanones, and heptenones (Figure 9). For example, Wijeratne isolated four new spirodecane compounds, oxaspirols A–D (**123**–**126**), from the endolichenic fungus *Parmotrema tinctorum* in 2016, which contained an intriguing furanone fragment with the formation of a spiro skeleton [46]. The furanone spirodecane oxaspirol B (**124**) exhibited inhibitory effects on p97 mutants and other ATP-related enzymes. In 2017, Kim et al. [47]. obtained two new furanones, grammicin (**127**) and patuli (**128**), from the endolichenic fungus *Xylaria grammica*. Compound **127** showed strong second-stage juvenile killing and egg-hatching-inhibitory effects, while **128** was strongly active towards various phytopathogenic bacteria in vitro.

In 2018, Ma et al. reported the isolation of six new 2-hydroxy-2-(1-hydroxyethyl)-2,3-dihydro-3(2*H*)-furanones, actinofuranones D–I (**129**–**134**), together with three known compounds, JBIR-108 (**135**), E-975 (**136**), and E-492 (**137**), from the endolichenic fungus *Streptomyces gramineus* [48]. Their anti-inflammatory activity tests proved that compounds **132**, **133**, **136**, and **137** could reduce NO production in a dose-dependent manner at varying concentrations of 15, 30, and 60 µM. In addition, compounds **132**, **133**, **136**, and **137** inhibited the LPS-induced release of proinflammatory cytokines interleukin-6 (IL-6) and tumor necrosis factor-*α* (TNF-*α*). Moreover, two new isobenzofuran-1(3H)-one derivatives, hypoxyolide A (**138**) and hypoxyolide B (**139**), were isolated from the endolichenic fungus *Hypoxylon fuscum* by Basnet et al. in 2019 [49], and hypoxyolide A (**138**) and hypoxyolide B (**139**) were firstly reported as new compounds. Interestingly, compound **138** showed moderate cytotoxic activity against the K562, SW480, and HEPG2 cell lines, with IC_50_ values ranging from 12.0 to 32.7 µM.

In 2015, Li et al. reported the isolation of two cyclopentenones, ophiosphaerekorrins A–B (**140**–**141**), from the crude extract of the endolichenic fungus *O. korrae*. Structurally, compounds **140**–**141** represented a naturally unprecedented chemical scaffold with a fascinating oxaspiro[4.4]nonenone substructure [50]. There were also five novel polyketides containing furanone fragments named javanicols A–E (**142**–**146**), together with two known compounds, (+)-terrein (**147**) and (−)-isoterrein (**148**), which were isolated from the fermentation broth of *Eupenicillium javanicum* in 2020 [51]. Further anti-inflammatory activity screening showed that javanicol E **146** and (+)-terrein **147** displayed moderate inhibitory effects on NO production, with IC_50_ values of 17.00 and 13.46 μM, respectively. In the same year, two closely related enantiomers with a novel 5/6-5 spiro skeleton from the endolichenic fungus *U. bilgramii* were discovered by Luan et al. and named (*R*)-ulodione A (**149a**) and (*S*)-ulodione A (**149b**) [52]. Their bioactivity evaluation demonstrated that these two compounds exhibited evident butyrylcholinesterase (BuChE)-inhibitory activity, with IC_50_ values of 9.0 ± 0.1 and 9.3 ± 0.2 μM, respectively.

In 2015, Zhao et al. isolated four new *α*-pyrone derivatives, nodulisporipyrones A–D (**150**–**153**), from an extract of *Nodulisporium* sp. [53]. In the same year, 11 *α*-pyrone derivatives, necpyrones A–E (**154**–**159**), PC-2 (**160**), (6*S*,10*S*)-LL-P880a (**161**), (6*S*,10*S*,20*R*)-LL-P880b (**162**), (1*S*,2*R*)-1-hydroxy-1-((*S*)-4-methoxy-6-oxo-3,6-dihydro-2H-pyran-2-yl)-pentan-2-yl acetate (**163**), and (*S*)-4-methoxy-6-pentanoyl-5,6-dihydro-2H-pyran-2-one (**164**), were obtained from the extract of the endolichenic fungus *Nectria* sp. by Li et al. [54]. Moreover, three similar *α*-pyrone derivatives named tolypocladones A–B (**165**–**166**) and 2H-pyran-2-one,4-methoxy-6-(1,3-pentadienyl) (**167**) were also discovered from the endolichenic fungus *Tolypocladium* sp. in 2017, as reported by Hu et al. [39]. Among them, compounds **165**–**166** were firstly reported as new *α*-pyrone derivatives.

Kim et al. discovered two naturally occurring novel *α*-pyrones, dothideopyrones E–F (**168**–**169**), from a culture of the endolichenic fungus *Dothideomycetes* sp. EL003334 in 2018 [55], and dothideopyrone F (**169**) had a considerable anti-inflammatory effect, with an IC_50_ value of 15.0 ± 2.8 μM, by inhibiting the expression of the iNOS and COX-2 proteins. Moreover, the first total syntheses of **168** and **169** were achieved by Aursnes et al. in 2022 [56]. In the same year, two new *δ*-lactones, talaromycin A (**170**) and clearanol A (**171**), were isolated from *Talaromyces* sp. [57]. Both of these compounds exhibited selective cytotoxicity against MDA-MB-231, with IC_50_ values of 24.6 ± 1.3 and 19.1 ± 1.2 μg/mL, respectively. Five naturally occurring *α*-pyrones, including the two new compounds, 5-*epi*-citreoviridin (**172**) and 5-*epi*-isocitreoviridin (**173**), together with three known *α*-pyrone derivatives, citreoviridin (**174**), isocitreoviridin (**175**), and aur-overtin U (**176**), were isolated from the fermentation broth of *E. javanicum* in 2020 [51]. In 2019, two new polyketides sharing an unusual furopyran-3,4-dione-fused heterocyclic skeleton and named ophiofuranones A–B (**177**–**178**) were isolated from the EtOAc extract of *O. korrae* by Li et al. [30].

In 2017, Yuan et al. reported the isolation and identification of five new polyketide–terpene hybrid metabolites (**179**–**183**) with highly functionalized groups, together with six known derivatives (**184**–**189**), from the endolichenic fungus *Pestalotiopsis* sp. [58]. After biological tests, compounds **179** and **183** were evidenced to exhibit pronounced antibacterial effects against *Fusarium oxysporum*, with an MIC value of 8 μg/mL. The biosynthetic pathway of compounds **179**–**189** was chemically postulated for the first time, which might pave the way for further biosynthesis research [58]. New polyketides, myxotritone C (**190**) and 7,8-dihydro-7*R*,8*S*-dihydroxy-3,7-dimethyl-2-benzopyran-6-one (**191**), were also isolated from the endolichenic fungus *Myxotrichum* sp. in 2016 by Yuan et al. [38], but neither of them showed any notable bioactivity.

Li et al. isolated and identified nine polyketide-derived compounds from the crude extract of *O. korrae* and assigned them with the trivial names of ophiosphaerellins A–I (**192**–**200**), and this lichen endophytic fungus was isolated from *Physcia* sp. [50]. Ophiosphaerellins A–I (**192**–**200**) shared a novel type of unprecedented scaffold with a bicyclo[4.1.0]heptenone backbone. In addition, the stereochemistry of C-1 in the bicyclo[4.1.0]heptenone ring system of ophiosphaerellins A-I (**192**–**200**) had been innovatively determined by considering the CEs at 210–240 nm as referring to π → π* transitions. Moreover, ophiosphaerellin C (**194**) exhibited obvious anti-ACH activity, with minimum inhibitory quantities of 1.25 μg/mL, which was weaker than that of galantamine (the positive control, 0.006 μg/mL).

Two polyketene compounds bearing a 10-member lactone micro-ring skeleton, named phomol (**201**) and 5,6-epoxy-phomol (**202**), were separated from the endolichenic fungus *H. fuscum by* Basnet et al. in 2019 [49]. Both compounds showed moderate cytotoxic activity against the K562, SW480, and HEPG2 cell lines, with IC_50_ values ranging from 12.0 to 32.7 µM. Moreover, phomol (**201**) displayed weak antibacterial activity against *S. aureus*, with an MIC value of 51.2 μΜ. In 2015, Samanthi et al. discovered a new microlide polyketide, 1,14-dihydroxy-6-methyl-6,7,8,9,10,10*α*,14,14*α*-octahydro-1H-benzo [f]oxacyclododecin-4(13H)-one (**203**), from the endolichenic fungus *C. trifolii* [37], and the biological evaluation of compound **203** found that it showed obvious radical scavenging activity, with an IC_50_ value of 4.0 ± 2.6 mg/mL, and exhibited a >90% inhibitory effect at 5 µg/mL towards the following five cancer cell lines: NCI-H460, MCF-7, SF-268, PC-3M, and MIA Pa Ca-2.

In 2018, two new macrolides, myxotrilactone A (**204**) and cladospolide B (**205**), were isolated from the extract of a solid-substrate culture of the endolichenic fungus *Myxotrichum* sp. by Yuan et al. [59]. Compound **204** displayed potent inhibitory activity on the root elongation of *Arabidopsis thaliana,* with more than a 75% inhibition rate at a concentration of 32 μg/mL. Similarly, compound **205** also displayed excellent inhibitory activity on the root elongation of *A. thaliana*, with a 69% inhibition rate at a concentration of 32 μg/mL. The aforementioned informative results collectively indicated that the endolichenic fungus *Myxotrichum* sp. might contribute to the defense capability of host lichens by producing phytotoxic secondary metabolites, which might also serve as ecologically beneficial and environmentally friendly mycoherbicides. Madyranga et al. also reported the isolation of a novel microlide compound, neurosporalol **206**, from the endolichenic fungus *N. ugadawe* in 2021, which was isolated from the lichen host *G. tsunodae* [44]. Neurosporalol **206** showed antioxidant activity, with an IC_50_ value of 5.03 ± 0.15 µg/mL, comparable to that of the positive control, BHT.

#### 2.1.3. Complex Aromatic Polyketides

Complex aromatic polyketides are usually constructed from two or more aromatic polyketide fragments. These types of natural products from endolichenic fungi are exemplified by naphthol polymers, xanthone dimers, perylenequinonoids, and spiciferone-derived dimers. Among them, naphthol polymers account for the largest proportion of the complex aromatic polyketides from endolichenic fungi, and the spirodioxynaphthalenes are well recognized as the typical structural types. The spirodioxynaphthalenes contain two 1,8-dihydroxynaphthalene-derived units with the formation of an interesting 6/6-6-6/6 spiroketal ring skeleton bridging through a typical spiroketal linkage, and most of them have been revealed to display a variety of biologically significant properties [60]. The main types of dinaphthalene compounds reported so far include spirodioxynaphthalenes with two oxygen bridges, dioxynaphthalenes with one oxygen bridge, spiromonoxynaphthalenes with a furan-sipro skeleton through one oxygen bridge and one C-C bridge, and fusodioxynaphthalenes with a new fuso-nonadiene skeleton.

Since 2015, 19 dimeric naphthalenes (Figure 10) have been discovered in endolichenic fungi. Among them, the main types include spirodioxynaphthalenes, dioxynaphthalenes, spiromonoxynaphthalenes, and fusodioxynaphthalenes. For example, Xie et al. reported four spirodioxynaphthalenes, palmarumycins P1–P4 (**207**–**210**), and three dioxynaphthalenes, juglanones C–E (**211**–**213**), from the endolichenic fungus *Phialocephala fortinii* for the first time in 2016 [61]. Among them, palmarumycin P3 (**209**) showed significant antifungal activity to reverse drug resistance, as well as weak cytotoxic activity against EC109 cells, with an IC_50_ value of 24.5 μM.

In 2019, three novel, uncommon types of spiromonoxynaphthalenes were observed to link the two naphthalene units together by one oxygen bridge (C4′-O-C4) and one C-C bridge (C4′–C6), and were named sacrosomycins A–C (**214**–**216**), which were obtained from the endolichenic fungus CGMCC3.15192 [62]. Besides these three novel spiromonoxynaphthalenes, there were three other known spiromonoxynaphthalene analogues, named urnucratin C (**217**), plecmillin A (**218**), and plecmillin C (**219**), which were also obtained from this endolichenic fungus. The anticancer effect tests showed that plecmillin A (**218**) exhibited the strongest cytotoxicity against HCT116, with an IC_50_ value of 2.1 μM. In 2022, Song et al. identified five undescribed perylenequinone fusodioxynaphthalenes, phialoce-phalarins H–L (**220**–**224**), together with two known fusodioxynaphthalenes, phialoce-phalarins A–B (**225**–**226**), from the endolichenic fungus *P. fortinii* [63]. Compounds **220**, **221**, and **225** showed weak cytotoxic activity against EC109 cells, with IC_50_ values ranging from 24.5 to 33.3 μM.

In addition, aromatic naphthalene dimers, trimers, and tetramers (Figure 11) have been found in the secondary metabolites of endolichenic fungi. Kim et al. obtained fourteen daldinol polyketides, including naphthol dimers, trimers, and tetramers, from *Daldinia childae* 047,219 by using a feature-based MS/MS molecular networking analysis in 2023 [45]. These aromatic polynaphthalenes mainly include six previously unreported naphthol tetramers: 1,1′,3′,3″,1″,1‴-quaternaphthalene-5,5′,5″,5‴-tetramethoxy-4,4′,4″4‴-tetraol (**227**), 1,1′,3′,1″,3″,3‴-quaternaphthalene-5,5′,5″,5‴-tetramethoxy-4,4′,4″4‴-tetraol (**228**), 3,3′,1′,1″,3″,3‴-quaternaphthalene-5,5′,5″,5‴-tetramethoxy-4,4′,4″4‴-tetraol (**229**), 1,1′,3′,3″,1″,3‴-quaternaphthalene-5,5′,5″,5‴-tetramethoxy-4,4′,4″4‴-tetraol (**230**), 3,1′,3′,1″,3″,3‴-quaternaphthalene-5,5′,5″,5‴-tetramethoxy-4,4′,4″4‴-tetraol (**231**), and 3,1′,3′,3″,1″,3‴-quaternaphthalene-5,5′,5″,5‴-tetramethoxy-4,4′,4″4‴-tetraol (**232**), and 8 known polyketides, daldinol (**233**), nodulisporin E (**234**), nodulisporin A (**235**), nodulisporin B (**236**), 1,1′,3′,3″-ternaphthalene-5,5′,5″-trimethoxy-4,4′,4″-triol (**237**), 3,1′,3′,3″-ternaphthalene-5,5′,5″-trimethoxy-4,4′,4″-triol (**238**), 1,1′,3′,1″-ternaphthalene-5,5′,5″-trimethoxy-4,4′,4″-triol (**239**), and 3,1′,3′,1″-ternaphthalene-5,5′,5″-trimethoxy-4,4′,4″-triol (**240**). These new naphthol tetramers are composed of four 5-methoxy-4-naphthol units, each connected to one another through a C-C single bond in various positions. Notably, compounds **235**–**237** demonstrated obvious anti-inflammatory activity.

In addition, 18 other complex aromatic polyketides (Figure 12) have been reported since 2015, and they include three xanthone dimers, six perylenequinonoids, and eight spiciferone-derived dimers. In 2019, Padhi et al. isolated four known secondary metabolites and identified them as aurasperone A (**241**), aurasperone D (**242**), asperpyrone A (**243**), and fonsecinone A (**244**), which were obtained from the culture of the endolichenic fungus *Aspergillus niger* [40]. In the same year, five perylenequinones identified as hypocrellin A (**245**), elsinochromes A–C (**246**–**248**), and phaeosphaerin C (**249**) were also isolated from the EtOAc extract of endolichenic fungus *O. korrae* by Li et al. [30]. The complex aromatic polyketide ES-242-3 (**250**) was also discovered from the endolichenic fungus *Talaromyces pinophilus* in 2022 by Shevkar et al. [34], and compound **250** showed strong antitumor activity on the MCF-7 and HeLa cell lines, with IC_50_ values of 14.08 ± 0.2 and 4.46 ± 0.05 μM, respectively.

In 2023, six spiciferone-derived dimers with an unprecedented skeleton, named phaeosphaerones A–F (**251**–**256**), were reported and isolated from the endolichenic fungus *Phaeosphaeria* sp. [64], and most of them were composed of a novel 4H-chromene-4,7(8H)-dione scaffold and an *α*-pyrone unit, forming fascinating dimeric compounds. Among them, compounds **251** and **253**–**256** possessed unconventional carbon skeletons featuring an ethylidene bridge, whereas compound **252** was a structurally interesting homodimer of spiciferone, which might be furnished by an unusual Rauhut–Currier reaction. Moreover, compounds **251**–**253** showed significant inhibitory effects towards the growth of dicot *Arabidopsis thaliana* at 100 μM. In addition, two biosynthetically related known secondary metabolites, lecanicillone A (**257**) and lecanicillolide (**258**), were also reported, and both showed very strong inhibitory effects on the fresh weight and root elongation of *Arabidopsis thaliana*, with IC_50_ values of 32.04 and 26.78 μM, respectively.

### 2.2. Polyphenyls

Polyphenyl derivatives are also commonly encountered in the secondary metabolites of endolichenic fungi, and most of them are terphenyls. Terphenyls are aromatic hydrocarbons consisting of one phenol core and two benzyl substituents. The terphenyl derivatives include *p*-terphenyls, *m*-terphenyls, and *o*-terphenyls, all of which usually show a wide range of biological activities. Among them, *p*-terphenyl derivatives are the most common natural terphenyls, in which two terminal benzene rings connect to the phenol mother ring at the *p*-position.

In recent years, more and more *p*-terphenyls have been found from endolichenic fungi (Figure 13). For example, Li et al. isolated ten new *p*-terphenyl derivatives, floricolins A–J (**259**–**268**), together with six known compounds, betulinan C (**269**), BTH-II0204-207 A (**270**), betulinan A (**271**), terphenyl 2 (**272**), 2-phenyl-[1H-2-benzopyran][4,3-e][*p*]-benzoquinone (**273**), and betulinan B (**274**), from the extract of the endolichenic fungus *Floricola striata* in 2016 [65]. Their biological activity assays showed that floricolins A–C (**259**–**261**) displayed moderate antifungal activity against *Candida albicans*, with MICs of 16, 8, and 8 μg/mL, respectively. Further antifungal mechanistic investigations revealed that the most active compound, compound **261**, exerted fungicidal action through the destruction of the cell membrane.

In 2018, Xu et al. disclosed the isolation and purification of eleven new *p*-terphenyls, floricolins K–U (**275**–**285**), and six biosynthetically related known terphenyl compounds (**286**–**291**) from the endolichenic fungus *F. striata*: kehokorin D (**286**), 3,5-diarylbenzoquinone (**287**), 2,4-dimethoxy-3,6-di(*p*-methoxyphenyl)phenol (**288**), 2′,3′,5′-trimethoxy-*p*-terphenyl (**289**), pentamethoxy-*p*-terphenyl (**290**), and terphenyl 3 (**291**) [66]. The cytotoxic activity results showed that compounds **279** and **280** exhibited significant cytotoxic activity against the A2780 cell line, with IC_50_ values of 3.4 and 8.6 μM, respectively. In the same year, two biphenyl compounds, 6′-methyl-[1,1′-biphenyl]-3,3′,4′,5-tetraol (**292**) and desmethylaltenusin (**293**), were isolated from *Talaromyces* sp. [57], and these two phenolic compounds displayed significant antioxidant activity, with an IC_50_ value equivalent to that of ascorbic acid. Two years later, Xie et al. isolated a diphenyl compound, ulophenol (**294**), from the endolichenic fungus *U. bilgramii* [24].

The biosynthetic pathway of terphenyl compounds originates from *L*-tyrosine, which undergoes an enzymatic amino transformation reaction to provide 4-hydroxylpheylpyruvic acid. The biosynthetic pathway of terphenylquinone is related to the critical precursor 4-hydroxyphenylpyruvic acid through the Claisen-type condensation reaction, and this conclusion was experimentally confirmed by ^13^C and ^14^C isotope labeling experiments. The biosynthesis of atromentin further demonstrated that the *L*-tyrosine was deaminated to 4-hydroxyphenylpyruvic acid by the PLP-dependent transaminase AtrD, which can transform the amino group into 2-oxoglutaric acid (2-OG). The further Claisen-type condensation of two molecules of 4-hydroxyphenylpyruvic acid to atromentin was catalyzed by the quinone synthetase AtrA (Figure 3) [67,68].

### 2.3. Terpenoids

Terpenoids are a series of structurally specific polymers constructed by isoprene units with different amounts and their structurally modified derivatives; their general formula can be theoretically considered as (C_5_H_8_)n. Terpenoids are broadly distributed in various plants, and they can also be frequently found in the secondary metabolites of microorganisms. Terpenoids are also commonly discovered in lichens’ endophytic fungi as one of their most major bioactive chemical constituents, and they show outstanding structural diversity, mainly composed of sesquiterpenoids, diterpenoids, and triterpenoids. The terpenoids isolated from lichens’ endophytic fungi usually exhibit a wide range of pharmaceutically significant bioactivities, with especially potent cytotoxic and antibacterial activity. Moreover, the terpenoids from endolichenic fungi can be further combined with sugar molecules to exist as glycosides. In this review, a total of 42 terpenoids obtained from endolichenic fungi since 2015 were collected and summarized, including 17 sesquiterpenes, 13 diterpenes (Figure 14), and 12 triterpenes together with three diterpene glycosides (Figure 15).

In 2015, Wu et al. [69] reported the isolation of a new cadinane-type sesquiterpene with the name of pericoterpenoid A (**295**) from the endolichenic fungal strain *Periconia* sp. No. 19-4-2-1. Compound **295** showed moderate antimicrobial activity against *A. niger* and weak activity against *C. albicans*. A new brasilane-type sesquiterpenoid glycoside named hypoxyside A (**296**) was also disclosed as a critical secondary metabolite in another endolichenic fungus in 2019 by Basnet et al. [49], and it showed obvious cytotoxic activity against K562, with an IC_50_ value of 18.7 μM. The sesquiterpene *rel*-(1*S*,4*S*,5*R*,7*R*,10*R*)-10-desmethyl-1-methyl-11-eudesmene (**297**) was also isolated from the endolichenic fungus *Daldinia childiae* for the first time as a new compound by Zhou et al. in 2020 [70] and exhibited notable *α*-amylase inhibitory activity. The sesquiterpenoid ophiokorrin (**298**) was first isolated and identified from the EtOAc extract of *O. korrae* by Li et al. in 2019 [30], and further phytotoxicity studies showed that ophiokorrin (**298**) could inhibit root elongation during germination, with an IC_50_ value of 18.06 µg/mL. In 2021, three novel sesquiterpenoids, sterpurol D (**299**), sterpurol E (**300**), and cryptomaraone (**301**), along with four known compounds, sterpurol A (**302**), sterpurol B (**303**), paneolilludinic acid (**304**), and murolane-2*α*,9*β*-diol-3-ene (**305**), were isolated from the lichen endophyte *Cryptomarasmius aucubae* [71]. Notably, compounds **299**, **300**, and **302**–**304** presented remarkable anti-inflammatory activity, with IC_50_ values ranging from 9.06 to 14.81 μM.

*Xylaria hypoxylon* is a lichen-associated fungus that was also found to have an outstanding capacity to produce novel secondary metabolites, and there were seven new bioactive eremophilane sesquiterpenes, eremoxylarins D–J (**306**–**312**), isolated from this fungus in 2023 [72]. The antimicrobial activity of eremoxylarins D–J (**306**–**312**) was evaluated, and the biological results showed that eremoxylarins D (**306**), F (**308**), G (**309**), and I (**311**) exhibited potent antibacterial activity against a series of Gram-positive bacteria (*S. aureus*, methicillin-resistant *S. aureus* (MRSA), and *S. epidermidis*), showing minimum inhibitory concentration (MIC) values ranging from 0.39 to 12.5 μg/mL. Among them, eremoxylarin I (**311**) was the most antibacterially active sesquiterpene, and it showed MIC values of 0.39–1.56 μg/mL, which were much greater than or similar to those of the positive control drug. Moreover, eremoxylarin I (**311**) was also biologically active against HCoV-229E, with an IC_50_ value of 18.1 μM, which was the concentration obtained without any noticeable toxicity to the hepatoma Huh-7 cell line.

In addition, pimarane diterpenoids have been broadly found in the secondary metabolites of endolichenic fungi, which can lead to a series of structurally diverse derivatives from the original core through enzymatic reactions, such as substitution, hydroxylation, acetylation, rearrangement, bromination, and ring expansion. In 2019, Hou et al. obtained three highly oxygenated pimarane diterpenoids, sarcosenones A–C (**313**–**315**), together with a known pimarane diterpenoid, 9*α*-hydroxy-1,8(14),15-isopimaratrien-3,7,11-trione (**316**), from cultures of *Sarcosomataceae* sp. [73], which is an endolichenic fungus found in the lichen *Everniastrum* sp. (*Parmeliaceae*). Furthermore, their biological evaluation illustrated that sarcosenone A (**313**) showed moderate cytotoxic activity towards a few cancer cell lines, with IC_50_ values of 7.5–26.4 μM. Moreover, Basnet et al. isolated two known diterpenoids, phaeropsidin A (**317**) and hymatoxin L (**318**), together with two reported diterpenoid glycosides, loxyisopimar-7-en-19-oic acid (**319**) and 16-*α*-D-glucopyrano-syloxyisopimar-7-en-19-oic acid (**320**), from the endolichenic fungus *Myrothecium inundatum* [74]. These diterpenoid compounds were found to exhibit cytotoxicity against the K562 and RKO human cancer cell lines, with IC_50_ values of 31.7–72.6 and 7.6–36.2 μM, respectively. Moreover, Basnet et al. discovered that the known isopimarane diterpene glycoside compound **319** displayed weak antibacterial activity against *S. aureus*, with an MIC value of 96.5 μΜ [49].

In 2022, Varlı et al. isolated three known diterpenoids, radianspenes C–D (**321**–**322**), and dahliane D (**323**), from the crude extract of *Nemania* sp. EL006872 ELF [75], and the three diterpenoids showed promising potential for innovative drug development for immune- and immuno-tumor therapies. Moreover, a previous pharmacological study revealed that radianspene C (**321**) had a significant cytotoxic effect on MDA-MB-435, with an IC_50_ value of 0.91 μM [76]. In 2023, Libertellenone T (**324**) was isolated also as a novel diterpenoid from the endolichenic fungus *Pseudoplectania* sp. (EL000327), and it exerted strong cytotoxicity towards the human CRC cell lines of Caco2, HCT116, DLD1, and HT29, with IC_50_ values of 17.5, 28, 36.6, and 28 µg/mL, respectively [77]. The investigation into its pharmacological mechanism confirmed that **324** could cause G2/M phase arrest and ROS/JNK signaling activation to induce the apoptosis of Caco2 cells. Moreover, **324** exhibited perfect synergistic effects in combination with various known and novel anticancer clinical agents.

In 2015, Li et al. isolated the tetracyclic triterpene helvolic acid (**325**) from the extract of endolichenic fungus *Nectria* sp. [54]. Moreover, six novel sesterterpenoids, asperunguisins A–F (**326**–**331**), and one closely related sesterterpenoid analogue, aspergilloxide (**332**), were isolated from the endolichenic fungus *Aspergillus unguis* (20141257a) in 2019 [78]. It should be noted that asperunguisins A–F (**326**–**331**) all featured a unique hydroxylated 7/6/6/5 tetracyclic system, and they are rarely occurring asperane-type sesterterpenoids. Biological assay tests showed that these sesterterpenoids displayed obvious cytotoxic activity. Among them, asperunguisin C (**330**) showed significant inhibitory activity towards A549, with an IC_50_ value of 6.2 μM. Further comprehensive studies of the relevant pharmacological mechanisms showed that **328** could cause DNA damage and induce G0/G1 phase arrest in tumor cells, thus leading to cancer cell apoptosis. In 2019, the chemical composition investigation of the endolichenic fungus *M. inundatum* also contributed to four new arborinane-type triterpenes and their related glycosides [74], including myrotheols A–B (**333**–**334**), together with the first two examples of natural 4-O-methyl-*α*-D-mannosides, myrothesides C–D (**335**–**336**). A biological activity evaluation showed that these compounds exhibited cytotoxicity against the K562 and RKO human cancer cell lines, with IC_50_ values of 28.6–63.9 and 51.0–68.8 μM, respectively.

### 2.4. Alkaloids and Their Biological Activities

Alkaloid compounds, with various unique structures or skeletons, are a group of basic organic compounds containing nitrogen atoms that widely exist in nature. Most of these compounds possess complex fused or bridged ring scaffolds, with the nitrogen atoms generally existing in the ring system. Numerous excellent medicinal research efforts have revealed that alkaloid compounds play significant and extensive roles in various biological processes, and they are recognized as an important research focus in the natural product, synthesis, biosynthesis, and medicine biochemical fields. Since 2015, a total of 52 alkaloids with nine cyclic peptide compounds have been found in the secondary metabolites of endolichenic fungi, and many of them have shown biologically meaningful activity (Figure 16). For example, Lee et al. successfully isolated three new diketopiperazines, cyclo(*L*-Pro-*D*-*trans*-Hyp) (**337**), cyclo(*L*-Pro-*D*-Glu) (**338**), and cyclo(*D*-Pro-*D*-Glu) (**339**), together with five known diketopiperazines, cyclo(*L*-Pro-*D*-*allo*-Thr) (**340**), cyclo(*L*-Pro-*L*-Asp)(**341**), cyclo(*D*-Pro-Gly) (**342**), cyclo(*L*-Pro-*L*-Ala) (**343**), and cyclo(*L*-Pro-*D*-Ala) (**344**), from the endolichenic fungus *Colpoma* sp. CR1465A in 2016 [79].

In 2019, a chlorinated isocoumarin and indole alkaloid hybrid metabolite named isocoumarindole A (**345**) was obtained from the endolichenic fungus *Aspergillus* sp. CPCC400810 by Chen et al. [80]. The novel alkaloid isocoumarindole A (**345**) was constructed with a chlorinated isocoumarin and an indoledione piperazine unit by the linkage of a carbon–carbon bond to form an unprecedented dimeric skeleton. The subsequent bioactivity test showed that isocoumarindole A (**345**) exhibited considerable cytotoxicity against the cancer cell lines of BT-549 and RKO, with IC_50_ values of 1.63 and 5.53 μM, respectively. In 2020, Padhi et al. isolated and identified a novel pyro-alkaloid, pyrophen (**346**), from the secondary metabolites of an endolichenic fungus, *Aspergillus niger* [40], which showed moderate antimicrobial activity, with an IC_50_ ranging from 35 to 97 mg/mL, against a broad spectrum of bacteria.

In the same year, a chemical investigation of the endophytic fungus *Xylaria psidii* from the lichen *Amandinea medusulina* led to the discovery of one new alkaloid compound, which was identified as (*Z*)-3-{(3-acetyl-2-hydroxyphenyl)diazenyl}-2,4-dihydroxybenzaldehyde (**347**) [50]. The bioactivity test showed that **347** had moderate cytotoxicity towards human lung cancer cells, with an IC_50_ value of 27.2 µg/mL. In 2021, Xu et al. reported the isolation of two pairs of diastereoisomeric isoindoline alkaloids, xylarins A–D (**348**–**351**), from the endolichenic fungus *Xylaria* sp., which was obtained from the lichen of *Parmelia* sp. [81]. Among them, compounds **348** and **349** possess an unprecedented skeleton comprising a complex 5/6/5-5/6 polycyclic ring system and exist as a pair of diastereoisomers, while another pair of diastereoisomers, diastereoisomers **350** and **351,** contain an additional *N*,*N*-dimethylaniline moiety at the C-3′ position. Interestingly, compound **351** exhibited obvious antithrombotic activity.

In addition to the new C7-alkylated salicylaldehyde derivatives, Lin et al. isolated nine known prenylated indole alkaloids, which were neoechinulin F (**352**), neoechinulin (**353**), neoechinulin C (**354**), tardioxopiperazine B (**355**), variecolorine O (**356**), cristatumin A (**357**), cryptoechinulin C (**358**), neoechinulin B (**359**), and 2-(2-methyl-3-en-2-yl)-1H-indole-3carbaldehyde (**360**), from the endolichenic fungus *A chevalieri* SQ-8 [42]. Among them, neoechinulin F (**352**) was firstly reported as a new compound. Moreover, the alkaloid neoechinulin C (**354**) had a significant anti-inflammatory effect in inhibiting nitric oxide production, with an IC_50_ value of 12 μM, and its possible anti-inflammatory mechanism was also studied via molecular docking. In 2022, Yuan et al. also isolated a diketopiperazine alkaloid, terreusinone (**361**), from *Talaromyces* sp., associated with *X. angustiphylla* [35].

Cyclopeptides are another type of important alkaloid existing in the endophytic fungi of lichens, and a few of them have been discovered and consequently reported in recent years. Since 2011, Wu has isolated two cyclic peptide alkaloids, cyclo(*N*-methyl-*L*-Phe-*L*-Val-*D*-Ile-*L*-Leu-*L*-Pro) and cyclo(*L*-Val-*D*-Ile-*L*-Leu-*L*-Pro-*D*-Leu), from the endolichenic fungus *Xylaria* sp. [82]. There were no cyclic peptides reported in the endophytic fungi of lichens for many years, but 10 cyclopeptides have been identified since 2015 (Figure 17). In 2021, seven new 3-hydroxy-4-methyldecanoic acid-containing cyclotetradepsipeptides, beauveamides A–G (**363**–**369**), and the known compound beauverolide Ka (**362**) were isolated from the endolichenic fungus *Beauveria* sp. by Zhou et al. [83]. Interestingly, all of them incorporate a 3-hydroxy-4-methyldecanoic acid (HMDA) moiety in their structures. Moreover, compounds **362**–**363** were disclosed to exhibit excellent protective effects on HEI-OC1 cells at 10 μM. While compounds **362**, **365**, and **366** could stimulate glucose uptake in cultured rat L6 myoblasts at 50 μM, compound **362** showed dose-dependent activity in both L6 myoblasts and myotubes. In 2022, Luo et al. also obtained a cyclic depsipeptide, xylaroamide A (**370**), from an endolicanic fungus *Xylaria* sp., via LC-MS-guided isolation [84]. Compound **370** showed significant cytotoxic activity on cell lines BT-549 and RKO, with IC_50_ values of 2.5 and 9.5 μM, respectively.

### 2.5. Steroids

Steroids are well known as one of the most critical chemical constituents participating in the construction of the cell membrane for various fungi, and they are also a group of natural compounds widely adopted as novel lead compounds in the medicinal industry. The structural types of steroids include phytosterols, bile acids, C21 steroids, cardiac glycosides, steroid saponins, and so on. Most steroid compounds share the basic skeletal structure of cyclopentanophenanthrene. In addition, there are usually two angularmethylgroups (C-10, C-13) and a side chain with different carbon atoms or oxygen-containing groups, such as hydroxyl and carbonyl groups (C-17), on the parent nucleus of cyclopentanophenanthrene. However, the number of steroids discovered from endolichenic fungi is very limited, as there are only 25 types of steroid compounds reported to date, with a total of 12 discovered since 2015 (Figure 18).

For example, Yuan et al. isolated two steroids, ergone (**389**) and ergosterol (**390**), from *Talaromyces* sp. associated with *X. angustiphylla* in 2018 [57]. Zhao et al. isolated eight new viridins, nodulisporiviridins A–H (**371**–**378**), from the extract of the endolichenic fungus *Nodulisporium* sp. in 2015 [85]. Among them, compounds **371**–**374** are a series of novel viridins with a unique ring-opened skeleton. Furthermore, the authors disclosed that nodulisporiviridin G (**377**) displayed potent A*β*42 aggregation-inhibitory activity, with an IC_50_ value of 1.2 μM, which then led to the discovery that **377** showed an outstanding capacity to significantly improve short-term memory, via a biological assay in an A*β* transgenic drosophila model for Alzheimer’s disease.

In the same year, they found another rare class of steroids (4-methyl-pro-gesteroids) in this fungus (*N.* sp.) via the OSMAC (one strain, many compounds) approach [86], and they named these ten new 4-methyl-progesteroid derivatives as nodulisporisteroids C–L (**379**–**388**). Interestingly, their biogenic synthesis (Figure 4) has also been chemically proposed. In the proposed biosynthetic pathway of compounds **379**–**390**, all of the fungal steroids are generally derived from lanosterol [87], and the demethylation of C-14 of lanosterol results in 4a-methyl-zymosterol, which is the critical precursor of compound **388**. Compound **381** can be generated by the Baeyer–Villiger oxidation of 4-methyl-8-en-pregnan-3,20-dione between the C-3 and C-4 positions, whereas compounds **382**–**387** are considered to be transformed from the intermediates **379** or **380** via chain opening and oxidative demethylation.

### 2.6. Others

In addition, there are a few other types (Figure 19) of bioactive secondary metabolites that have been discovered from endolichenic fungi, such as fatty acids, chain alcohols, etc. Wang et al. isolated two small molecules of (*E*,*E*)-4-hydroxymethyl-4,6-octadien-2,3-diol (**391**) and lachnellin B (**392**) from the endolichenic fungus *A. montagnei* in 2017 [23]. In 2019, a new polyacetylene glycoside, fuscuyne (**393**), was isolated from the solid culture of the endolichenic fungus *H. fuscum* by Basnet et al. [49]. Zhou et al. also discovered two new alkanols, 2,4,5-heptanetriol (**394**) and 6-heptene-2,4,5-triol (**395**), from the endolichenic fungus *Daldinia childiae* in 2020 [70]. In the same year, Tan et al. obtained acetyl tributyl citrate (**396**) from the ethyl acetate extract of *Fusarium proliferatum* [88], and biological activity tests showed that **396** exhibited significant antibacterial activity against *Klebsiella pneumoniae*, *Pseudomonas aeruginosa*, and *S. aureus*, with IC_50_ values of 3.11, 0.19, and 0.78 μM, respectively.

In 2021, (–)-10,11-dihydroxyfarnesol (**397**), which showed significant anti-inflammatory activity with an IC_50_ value of 9.06 μM, was isolated from the lichen endophyte *C. aucubae* [71]. The novel oxygenated fatty acid (8*Z*)-5,6-epoxy-4-hydroxy-octadec-8-enoic acid (**398**), with anti-*Candida albicans* biofilm activity, was also isolated from the crude extract of *Preussia persica* by Toure et al. in 2022 [89]. Moreover, in 2023, Varlı et al. found that the fatty acid subfractions of the endolichenic fungus *Phoma* sp. EL006848 could efficiently suppress multiple immune checkpoints via the inhibition of the benzo[a]pyrene-induced (an AhR ligand) expression of PD-L1 [90], and a further extensive phytochemical investigation successfully led to the isolation and identification of palmitic acid (**399**), stearic acid (**400**), and oleic acid (**401**), thus unambiguously verifying that the fatty acids from the endolichenic fungi *Phoma* sp. possessed strong potential for immunotherapy.

## 3. A Meta-Analysis of Research Progress and Status of Endolichenic Fungi

### 3.1. The Main Research Groups Engaged in the Study of Endolichenic Fungi

During the summary of chemical composition and biological activity, a comprehensive analysis of the research on the secondary metabolites of endolichenic fungi from 2015 to the present was conducted, as shown in Figure 20. It unambiguously shows that the number of published articles worldwide is very stable and is maintained at approximately 10 articles per year. Meanwhile, the total number of compounds reported annually fluctuates slightly, usually maintained at a rate of between 40 and 80 compounds for each year. In particular, 20–40 new compounds have been discovered in the field of natural product chemistry every year since 2015.

Based on the research articles published in internationally renowned journals on the secondary metabolites of lichen endophytic fungi since 2015, the main research groups contributing to the natural product chemistry of lichen endophytic fungi are also preliminarily summarized. As shown in Figure 21, it can be found that the most active research groups related to lichen endophytic fungi are those of Hongxiang Lou, Hanggun Kim, Hao Gao, and Priyani A. Parargama. These four research groups have contributed more than a half of all published research articles from 2015 to the present.

Moreover, the total number of isolated compounds and new ones discovered from endolichenic fungi by the aforementioned main research groups are tentatively analyzed (Figure 22). Hongxiang Lou’s research group has made a major contribution to the discovery of secondary metabolites from endophytic fungi in lichen. As illustrated in Figure 22, a total of 215 natural products have been reported by this group, and there are 124 natural products that have been reported as new compounds. Interestingly, the total number of natural products and new compounds reported by Hongxiang Lou’s research group accounts for more than a half of all the secondary metabolites isolated from endolichenic fungi.

At the same time, according to the research articles on the secondary metabolites of lichen endophytic fungi published in internationally renowned journals since 2015, the main research area distribution of natural products of lichen endophytic fungi is further summarized. As shown in Figure 23, it can be found that research related to endolichenic fungi is mainly distributed in Asia, with a slight level of engagement in Europe and North America. Since 2015, China has contributed more than half of the research papers on endophytic fungi in lichens, followed by South Korea with 18%. Therefore, it is not difficult to find that the research of endophytic fungi in lichen is mainly concentrated in Asia, while there is less research being conducted in other continents, which also indicates that a large number of endophytic fungi in lichen need to be explored.

### 3.2. The Fungal Sources and Structural Characteristics of the Isolated Secondary Metabolites

The endolichenic fungi obtained from lichens are strategically important bioresources for the microbial and medicinal industries. Thus, the origination and species of endolichenic fungi are also comprehensively analyzed in this review, in order to provide detailed and clear information for potential readers to facilitate a better understanding. A species analysis of the endolichenic fungi reported in recent years is illustrated in Figure 24, and it can be readily found that 35 endolichenic fungi from 20 species of lichens in five districts have been investigated since 2015. Among these endolichenic fungi, *Xylaria* sp., *Aspergillus* sp., *Nodulisporium* sp., *Daldinia* sp., and *Talaromyces* sp. collectively represent the predominant endophytes with the largest proportions and that are the most frequently studied.

With the further aim of providing an unambiguous depiction of the secondary metabolites generated by the major strains of endolichenic fungi, a careful analysis of the critical chemical compositions of the major strains is also performed, and the informative results are depicted in Figure 25. From the statistical data, it is easily found that the secondary metabolites from most of the endolichenic fungi are mainly polyketides, with the exception of the fungi of *Xylaria* sp., *Aspergillus* sp., and *Nodulisporium* sp. Moreover, the structural types of the secondary metabolites from these three strains are relatively diverse, evenly distributed as polyketides, terpenoids, and alkaloids. Interestingly, a series of novel compounds with unprecedented skeletons have been found in the four endolichenic fungal strains of *Xylaria* sp., *Aspergillus* sp., *Daldinia* sp., and *Phaeosphaeria* sp.

### 3.3. The Biological Activity of the Secondary Metabolites of Endolichenic Fungi

The biological activity of secondary metabolites is an important subject for natural product and medicinal chemists. In this review, the biological activity of the secondary metabolites discovered from endolichenic fungi since 2015 is also extensively summarized and carefully analyzed (Table 1, Table 2, Table 3, Table 4 and Table 5). According to the statistical data, more than 40 biologically meaningful natural products have been isolated and pharmacologically evaluated since 2015, and their biological properties include cytotoxic, antimicrobial, anti-inflammatory, antioxidative, anti-influenza A virus, and other related biological effects. The structural types of the reported compounds and their biological activity are comprehensively analyzed, and the conclusive results are shown in Figure 26. Notably, these results collectively indicate that the rich and significant biological activity of secondary metabolites from endolichenic fungi might be closely related to their structural diversity. For example, a larger number and various structural types of polyketides are included in the secondary metabolites from endolichenic fungi, and additional types of bioactive molecules remain to be discovered.

Finally, the bioactivity of the secondary metabolites of the most prominent strains is statistically analyzed (Figure 27), and it is found that the bioactivity of different strains is also significantly different, such as that of the endolichenic fungi *Xylaria* sp. The secondary metabolites mainly reflect the cytotoxicity and antibacterial activity of the strain *Aspergillus* sp. It mainly shows cytotoxicity, antimicrobial activity, and anti-inflammatory effects. Meanwhile, the strain *Daldinia* sp. shows antioxidant and antiviral activities.

## 4. Conclusions and Outlook

Lichens are naturally occurring as stable microorganism-like plants with complex compositions of known ascomycetes, photosynthesizing partners, and specific basidiomycete yeasts. Because of their extremely unique growth environments, they are widely acknowledged as impressive microorganisms with an outstanding capacity to produce a variety of biologically significant and structurally fascinating secondary metabolites with rich structural diversities. Endophytic fungi, as the main components of lichens, can also produce secondary metabolites with novel structures and extensive bioactivities based on their mutualistic symbiosis with their hosts, which have thus made them appealing targets and attracted extensive research attention from natural product and pharmaceutical chemists. In recent years, many research groups have been motivated to conduct research on the discovery of novel secondary metabolites from endolichenic fungi. According to the number of isolated compounds and the research articles published in renowned journals in recent years, it can be readily found that research efforts and contributions with regard to the secondary metabolites of endophytic fungi in lichens have steadily increased, and they collectively demonstrate endolichenic fungi’s increasing importance in the field of natural product chemistry.

According to the careful inspection and review of research efforts made since 2015, a variety of endophytic fungi were obtained from lichens by surface sterilization, and a number of species of endolichenic fungal strains with broad biological activities and high chemical abundance were selected as targeted strains for further extensive chemical studies, to obtain promising lead compounds with biologically significant activities and potential economic value. Thus, the urgent requirement for innovative drug development in the modern pharmaceutical industry can be satisfied. The specific process followed by natural product chemists in studying endolichenic fungi is introduced in Figure 28. Among past research achievements, a total of 583 compounds have been successfully isolated and identified from endolichenic fungi, of which 374 have been introduced as new compounds. Since 2015, 35 species of lichen endophytic fungi have been extensively investigated with the aim to discover biologically active or structurally novel natural products. As a result, a total of 407 compounds, with 270 new ones, have been reported, which include 16 novel compounds with unprecedented skeletons. Most importantly, more than 30 active lead compounds have been described to show significant potential in new drug development.

After a comprehensive analysis of the research status of this topic since 2015, it is easily found that the main secondary metabolites in endolichenic fungi share outstanding characteristics in terms of structural diversity, and they usually contain polyketides as their major bioactive chemical components, followed by alkaloids, terpenes, steroids, and so on. The great structural diversity of the secondary metabolites in endolichenic fungi may also contribute to the diversity of their biological activity. Through careful inspection and a statistical analysis, it is found that the secondary metabolites of lichen endophytic fungi show extensive biological activities, including not only significant cytotoxic, antioxidant, and anti-inflammatory activities, but also noticeable antiviral and root-growth-inhibitory biofunctions. In summary, the numerous excellent studies on the secondary metabolites of endophytic fungi in lichens collectively indicate that they are emerging as a research hotspot for both natural product and pharmaceutical chemists. Moreover, it is believed that further scientific research on the secondary metabolites of endolichenic fungi will be beneficial to enrich the compound library of chemical molecules, as well as improve the biological activities of drug molecules.

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
