# Peer review of "Endolichenic Fungi: A Promising Medicinal Microbial Resource to Discover Bioactive Natural Molecules—An Update"

_jof, 2024, doi:10.3390/jof10020099_

Round 1

Reviewer 1 Report

Comments and Suggestions for Authors

As a general remark, to improve the review, I suggest that the authors discuss the influence of culture media in the production of metabolites. For example, glycosides metabolites are not so frequent: are they related to a specific culture medium? It could be interesting to add a paragraph (3.4. for example) dedicated to the culture media commonly used to isolate endolichenic metabolites.

In the paragraph 3.1, page 33, the author summarized the main research groups engaged in the study of endolichenic fungi. My question is: are the researches concentrated in some continents or country? This could suggest that an important quantity of endolichenic fungi is still unexplored and understudied. This could be discussed in the conclusion for example.

Please also take into consideration the following remarks and suggestions.

-          Title: why the term “microresource”? Is this term really appropriate since endolichenic fungi can be cultivated and obtained in high amount?

-          In the abstract: please modified the term “lower plants” line 9 since fungi are different from plants. The same with the term “plant-like” line 10.

-          Line 37-38: I don’t understand “ascomycetes are the pivotal symbiotic bacteria”

-          Line 39: replace “symbiotic algae” by “symbiotic partner”

-          Line 136-138: please rephrase to be more understandable.

-          Line 162: put this phrase after “diversity” line 166.

-          Figure 2.2: separate the anthraquinone from the other quinones to follow the text.

-          Line 332: Add a space between “Aspergillus” and “niger” and suppressed “Tiegh”

-          Line 308: suppressed “4259a”

-          Line 313: suppressed “EL002650”

-          Figure 2.10. Some structure can be grouped, for example 222, 223 and 224.

Figure 3.7. In the legend, “anti-influenza A” corresponds to an antiviral activity. It could be interesting to separate antifungal and antibacterial activity instead of antimicrobial.

Comments on the Quality of English Language

The english language is good even if some phrases could clearer.

Author Response

1. Point-by-point response to Comments and Suggestions for Authors

Comments 1: As a general remark, to improve the review, I suggest that the authors discuss the influence of culture media in the production of metabolites. For example, glycosides metabolites are not so frequent: are they related to a specific culture medium? It could be interesting to add a paragraph (3.4. for example) dedicated to the culture media commonly used to isolate endolichenic metabolites.

Response 1:

Thank you for your careful review, and we think your proposal is very good, but in the current research process of endolichenic fungi, rice medium is mainly used for large fermentation, and good rules and conclusions cannot be drawn for the time being. However, we can make more attempts on your proposal in our subsequent research.

Comments 2: In the paragraph 3.1, page 33, the author summarized the main research groups engaged in the study of endolichenic fungi. My question is: are the researches concentrated in some continents or country? This could suggest that an important quantity of endolichenic fungi is still unexplored and understudied. This could be discussed in the conclusion for example.

Response 2:

We highly appreciated your careful review and constructive comment. we have re-analyzed the data and added this part to the article, for details, see page 32, line 990.

Comments 3: Title: why the term “microresource”? Is this term really appropriate since endolichenic fungi can be cultivated and obtained in high amount?

Response 3:

Thank you for your careful review, and we have corrected “microresource” to “microbial resources” with your helpful suggestion.

Comments 4: In the abstract: please modified the term “lower plants” line 9 since fungi are different from plants. The same with the term “plant-like” line 10.

Response 4: Thank you very much for your constructive suggestion. We have rechecked and proofread the concept, and the details can be found in page 1 and line 10.

Comments 5: 2. Line 37-38: I don’t understand “ascomycetes are the pivotal symbiotic bacteria”

Response 5: Thank you for your careful review, and we have corrected “bacterial” to “fungi” with your helpful suggestion.

Comments 6: Line 39: replace “symbiotic algae” by “symbiotic partner”

Response 6: Thank you for pointing out this mistake. We have changed “symbiotic algae” to “symbiotic partner” as you suggested.

Comments 7: Line 136-138: please rephrase to be more understandable.

Response 7: We highly appreciated your careful review and kind reminder. We have changed the sentence “Great chemical efforts devoted to the discovery of pharmacologically meaningful secondary metabolites of ELF have emerged.” to “In recent years, the chemical efforts devoted to the discovery of pharmacologically meaningful secondary metabolites of ELF have massively emerged and conducted.”

Comments 8: Line 162: put this phrase after “diversity” line 166.

Response 8: Sincerely thank you for your suggestions. We have modified and adjusted the logic and statements in this section.

Comments 9: Figure 2.2: separate the anthraquinone from the other quinones to follow the text.

Response 9: We highly appreciated your careful review and kind reminder. We have separated the anthraquinone from the other quinones as you suggested.

Comments 10: Line 332: Add a space between “Aspergillus” and “niger” and suppressed “Tiegh”

Response 10: Thank you for pointing out this mistake. We have changed “Aspergillus niger Tiegh” to “Aspergillus niger”.

Comments 11: Line 308: suppressed “4259a”

Response 11: We sincerely thank you for your careful review and informative comment. And, we have deleted “4259a”.

Comments 12: Line 313: suppressed “EL002650”

Response 12: We are highly grateful for your constructive suggestion, and we have deleted “EL002650”.

Comments 13: Figure 2.10. Some structure can be grouped, for example 222, 223 and 224.

Response 13: We sincerely thank you for your careful review and kind reminder! We have regrouped this section to make it much clearer for readers to understand.

Comments 14: Figure 3.7. In the legend, “anti-influenza A” corresponds to an antiviral activity. It could be interesting to separate antifungal and antibacterial activity instead of antimicrobial.

Response 14: We highly appreciated for your kind reminder and constructive suggestion. We have readjusted that section and separated antimicrobial activity to antifungal and antibacterial activity as you suggested.

2. Response to Comments on the Quality of English Language

Point 1: The English language is good even if some phrases could clearer.

Response 1: Thank you for your suggestions. We have checked our manuscript and carefully revised the spelling and grammar errors in our revised manuscript with the aids of the reviewers and another professional natural product chemist accordingly. In addition, the English language of this manuscript was polished by professional editors from Multidisciplinary Digital Publishing Institute.

Reviewer 2 Report

Comments and Suggestions for Authors

The contents of this manuscript were within the scope of J. Fungi and were well prepared and are therefore recommended for publication.

Several minor suggestions

 L 37-38 Ascomycetes are the pivotal symbiotic bacteria that account for their bacterial diversity.

Ascomycetes are the fungi!

Figure 3.4. The main types of endolichenic fungi and their proportion.

The captions around the diagram are not clear. Better remove it. Enlarge the legend in the color scheme and indicate the percentage of occurrence next to the name of the fungi.

2. Different Types of Natural Products from Endolichenic Fungi

In this section you describe not only the structure of compounds but also biological activity! Indicate this in the section title and then move tables 1-5 to this section and provide a link to these tables in the text.

Or describe only structures in this section.

polyketide 203 - biological activity is not indicated in the table, but is given in the text.

Author Response

Comments 1: L 37-38 Ascomycetes are the pivotal symbiotic bacteria that account for their bacterial diversity. Ascomycetes are the fungi!

Response 1: Thank you for your careful review, and we have corrected “bacterial” to “fungi” with your helpful suggestion.

Comments 2: Figure 3.4. The main types of endolichenic fungi and their proportion. The captions around the diagram are not clear. Better remove it. Enlarge the legend in the color scheme and indicate the percentage of occurrence next to the name of the fungi.

Response 2: Thank you very much for your constructive suggestion. We have recalibrated Figure 3.4 and listed their proportion of different strains, and the details can be found in page 37 and line 900.

Comments 3: 2. Different Types of Natural Products from Endolichenic Fungi

In this section, you describe not only the structure of compounds but also biological activity! Indicate this in the section title and then move tables 1-5 to this section and provide a link to these tables in the text. Or describe only structures in this section.

Response 3: We sincerely thank for careful review and valuable suggestion. Although the biological activities of the summarized compounds are also introduced in part 2, the part is still focused to highlight the structural diversity. Moreover, the summary of the biological activity in tables 1-5 is presented as a whole analysis for this review, so we have placed them in the part 3 for readers as a summarized information.

Comments 4: polyketide 203 - biological activity is not indicated in the table, but is given in the text.

Response 4: We highly appreciated your careful review and kind reminder. We have carefully checked this section, and found that the antioxidant activity data of polyketide 203 were summarized in the table. The antitumor activity was not shown in the table, because only the inhibition rate of polyketide 203 was reported in the literature. Therefore, we only summarized the antioxidant activity data of polyketide 203 in this manuscript.

Reviewer 3 Report

Comments and Suggestions for Authors

The review shows a comprehensive and systematic evaluation of the secondary metabolites of endolichenic fungi. The origin, distribution, structural characteristics of natural products, as well as, their biological activity and recent advances in the medicinal application were summarized since 2015.   Some minor improvements described below should be done before the publication.

Line 37:  ascomycetes are not bacteria.  This data should be corrected.

Line 47: the citation of references should be corrected.

Figures 2.1 & 3.5.:  steroids instead sterides

Figure 2.2.:  the structures 17a and 17b are not enantiomers as they were represented.  The pair of structures should be revised.

Line 228: precursor instead procursor.

Scheme 2: lactonization instead lactainzation

Lines 310 & 311:  the hydroxy word does not need a hyphen.  Verify it !

Line 320: the dimethyl word does not need a hyphen. The name should be corrected.

Lines 332, 563 & 750:  Aspergillus niger Tiegh.   The use of italic form should be revised.

Line 399: actinofuranone D-I instead actinofuranonea D-I

Line 413:  In 2015, Li et al. reported the isolation of two biosynthetic relatives....   please check the phrase.   

Lines 802-803: keratyl functional groups?  Please clarify it !

Line 810:  endolichenic fungus instead endolichenic fugue.

Figures 3.2. & 3.3.:  from my point of view, this analysis is not relevant in the revision.   This data is already implicit in the cited references.

Figure 3.4.: A lot of fungi names are difficult to understand given the quality of the figure.    Perhaps the 10 most relevant fungi could be prioritized in the figure and the current figure moved to the supplementary material.

Comments on the Quality of English Language

Minor editing of English language is required.  These corrections i.e. typo errors, incorrect names, etc... were summarized above.

Author Response

1. Point-by-point response to Comments and Suggestions for Authors

Comments 1: Line 37: ascomycetes are not bacteria. This data should be corrected.

Response 1: We highly appreciated for your careful review and informative comments. We have corrected “bacterial” to “fungi” with your helpful suggestion.

Comments 2: Line 47: the citation of references should be corrected.

Response 2: We are so sorry for the careless mistake, and we have carefully checked and corrected this problem.

Comments 3: Figures 2.1 & 3.5.: steroids instead sterides.

Response 3: We are sincerely sorry for our carelessness, and we have corrected “sterides” to “steroids” with your helpful suggestion.

Comments 4: Figure 2.2.: the structures 17a and 17b are not enantiomers as they were represented.  The pair of structures should be revised.

Response 4: We are sincerely thank you for your careful review and informative suggestion, and we have carefully checked and corrected it in our revised manuscript.

Comments 5: Line 228: precursor instead procursor.

Response 5: Sincerely thank you for your suggestions. We have corrected “procursor” to “precursor”.

Comments 6: Scheme 2: lactonization instead lactainzation

Response 6: We highly appreciated your careful review and kind reminder. We have corrected “lactainzation” to “lactonization” following your suggestion.

Comments 7: Lines 310 & 311: the hydroxy word does not need a hyphen. Verify it!

Response 7: We are sincerely sorry for our carelessness, and we have checked and corrected it with the aid of your constructive suggestion.

Comments 8: Line 320: the dimethyl word does not need a hyphen. The name should be corrected.

Response 8: We sincerely thank you for your careful review and kind reminder! We have carefully corrected this problem.

Comments 9: Lines 332, 563 & 750:  Aspergillus niger Tiegh.   The use of italic form should be revised.

Response 9: Thank you for pointing out this mistake. We have changed “Aspergillus niger Tiegh” to “Aspergillus niger Tiegh” as you suggested.

Comments 10: Line 399: actinofuranone D-I instead actinofuranonea D-I

Response 10: Thank you very much for your careful review. We have changed “actinofuranonea D-I” to “actinofuranone D-I” in our revised manuscript.

Comments 11: Line 413:  In 2015, Li et al. reported the isolation of two biosynthetic relatives.... please check the phrase.

Response 11: We are highly grateful for your careful review and constructive suggestion! We have changed the sentence “In 2015, Li et al. reported the isolation of two biosynthetic relatives....…” to “In 2015, Li et al. reported the isolation of two cyclopentenones, ophiosphaerekorrins A-B …” with the aid of your suggestion.

Comments 12: Lines 802-803: keratyl functional groups?  Please clarify it!

Response 12: Thank you very much for your careful review. We have changed “keratyl functional groups” to “angular methyl groups”.

Comments 13: Lines 810: endolichenic fungus instead endolichenic fugue.

Response 13: Many thanks for your suggestion. We have changed “endolichenic fugue” to “endolichenic fungus”.

Comments 14: Figures 3.2. & 3.3.: from my point of view, this analysis is not relevant in the revision. This data is already implicit in the cited references.

Response 14: We highly appreciated for your careful review and informative comments. Although this data of Figures 3.2. and 3.3 are already implicit in the cited references, the chart forms can be much more direct and better to provide clear information of the current research status of endolichenic fungi for the readers than that of the cited literatures.

Comments 15: Figure 3.4.: A lot of fungi names are difficult to understand given the quality of the figure. Perhaps the 10 most relevant fungi could be prioritized in the figure and the current figure moved to the supplementary material.

Response 15: We sincerely thank you for your careful review and kind reminder! We have carefully readjusted the chart to make it much clearer for readers to understand.

2. Response to Comments on the Quality of English Language

Point 1: Minor editing of English language is required. These corrections i.e. typo errors, incorrect names, etc... were summarized above.

Response 1: Thank you for your suggestions. We have checked our manuscript and carefully revised the spelling and grammar errors in our revised manuscript with the aids of the reviewers and another professional natural product chemist accordingly. In addition, the English language of this manuscript was polished by professional editors from Multidisciplinary Digital Publishing Institute.
